



# Contribution of Asian emissions to upper tropospheric CO over the remote Pacific

Linda Smoydzin[1] and Peter Hoor[1]

[1]Institute for Atmospheric Physics, Johannes Gutenberg University, Mainz, Germany

**Correspondence:** L. Smoydzin (smoydzin@uni-mainz.de)

**Abstract.** We use CO data from the MOPITT satellite instrument from 2000-2019 to compose a climatology of severe pollution events in the mid- and upper troposphere over the northern-hemispheric (NH-) Pacific. To link each individual pollution event detected by MOPITT with a CO source region, we performed trajectory calculations using MPTRAC, a lagrangian transport model. To analyse transport pathways and uplift mechanisms we combine MOPITT data, the trajectory calculations and ERA-Interim reanalysis data.

Events of elevated CO which we detect at level between 500hPa and 300hPa over the NH-Pacific throughout the year, occur with a surprisingly high regularity and frequency (70% of all days during winter, 80% respectively during spring). Our study further indicates, that the spatial coverage of individual upper tropospheric pollution cluster increased in spring time during the 20 years we analysed.

The position of upper tropospheric pollution plumes show a strong seasonal cycle. During winter, most pollution events are detected over the north-eastern and central NH-Pacific, during spring over the central NH-Pacific and during summer over the western NH-Pacific. We detect most pollution episodes during spring. Trajectory simulations reveal China as the major CO-source region throughout the year. The contribution of other source regions shows a strong seasonal cycle: NE-Asia is a significant CO-source region during winter and summer while India and SE-Asia are important source regions mainly during spring.

## 1 Introduction

The long-range transport of trace gases and aerosols from East Asia across the Pacific has been subject of investigation for many years as pollution plumes can be lifted to the free troposphere where they are quickly transported to the North Pacific and North American west coast by midlatitude storm tracks (Yienger et al., 2000; Liang et al., 2004; Holzer et al., 2005; Liang et al., 2005; Wuebbles et al., 2007; Turquety et al., 2008). A lot of effort has been spent in the last years to quantify the contribution of trace gases and aerosols of Asian origin to north American pollution levels (e.g. Zhang et al., 2008; Yu et al., 2008; Hu et al., 2019; Yu et al., 2019). Though these studies mainly focus on the chemical processing of the pollution plume rather than dynamical transport aspects. Several studies use satellite derived CO data to better understand vertical and horizontal transport mechanisms in relation to the prevailing synoptic situation. Ding et al. (2015) use data from numerical models and the MOPITT satellite instrument to investigate high levels of CO in the upper troposphere over the eastern Pacific





while Liu et al. (2006) examine the influence of synoptic processes on the distribution of tropospheric CO also using MOPITT satellite data.

Many studies in recent years focused on the role of the Asian summer monsoon anticyclone for export of trace gases, pollutants or aerosols from Asia to the stratosphere or into the subtropical upper troposphere and tropical tropopause layer (e.g. Garny and Randel, 2016; Müller et al., 2016; Vogel et al., 2016; Yu et al., 2017; Santee et al., 2017; Lelieveld et al., 2018). Export of Asian pollution to the upper troposphere of mid and high latitudes however occurs at lower levels during the whole year. Numerous studies reveal a seasonal maximum of the Asian pollution outflow in spring. Studies, using CO as a pollution tracer explain this observation with efficient ventilation of the Asian boundary layer via midlatitude cyclones and convection and increasing biomass burning emissions compared to winter (Liang et al., 2004; Holzer et al., 2005; Zhang et al., 2008; Yu et al., 2008; Luan and Jaeglé, 2013). Even though the strength and frequency of such pollution outflow events vary with season, they occur regularly throughout the year (Liang et al., 2004; Han et al., 2018). As stated by e.g. Liang et al. (2004), midlatitude cyclones seem to be the driving mechanism for long range transport of polluted air masses from Asia to the western Pacific or north American west coast. More than half a century ago first studies reveal the existence of air streams such as the warm conveyor belt (wcb), cold conveyor belt and dry intrusion associated with extratrocpical cyclones. Since then, several studies investigate dynamical aspects of wcb's (e.g. Wernli and Davies, 1997; Wernli, 1997; Madonna et al., 2014; Gehring et al., 2020) as well as vertical transport of polluted air masses in warm conveyor belts in particular over the northern Pacific (e.g. Fuelberg et al., 2003; Kiley and Fuelberg, 2006; Klich and Fuelberg, 2014; Dickerson et al., 2007; Heald et al., 2003). Apart from uplift in the vicinity of a cyclone, orographic lifting (Chen et al., 2009; Ding et al., 2009) and convection plays a major role for vertical transport of Asian pollution plumes especially during summer (Bey et al., 2001; Kiley and Fuelberg, 2006; Liang et al., 2005, 2007, 2004; Vogel et al., 2014).

Even though wcb's occur more frequent in the northern hemispheric winter than summer (Eckhardt et al., 2004; Madonna et al., 2014), their contribution to the uplift of polluted air masses is presumably important throughout the year (Liang et al., 2004). Madonna et al. (2014) find a global maximum of wcb's over the western North Pacific, China, and Taiwan during NH summer. They discuss in detail that presumably a nearly stationary low-level baroclinic zone (Yihui and Chan, 2005; Ninomiya and Shibagaki, 2007) together with the East Asian monsoon, is responsible for tropical moisture exports over the west Pacific in summer explaining the high frequency of wcb's during June, July and August (Knippertz and Wernli, 2010). Generally, the Asian monsoon controls directly or indirectly the pollution outflow to the Pacific throughout the year (e.g. Kaneyasu et al., 2000; Müller et al., 2016; Lelieveld et al., 2018; Tomsche et al., 2019).

Many studies addressing long range transport of Asian pollution to the western Pacific or Arctic are based on case studies (e.g. Heald et al., 2003; Di Pierro et al., 2011; Matsui et al., 2011; Roiger et al., 2011). Several of those studies see uplift in the vicinity of a cyclone as an important part of the pollution transport process though, those studies show very different, individual transport pathways across the Pacific. Luan and Jaeglé (2013) present a composite analysis of aerosol export events from Asia to north America. Based on their analysis using MODIS satellite data and the GEOS-CHEM model, they find an enhancement of AOD values over the north-eastern Pacific/Alaska and over the south-western Pacific. The two separated maxima in the AOD composites appear as a split of the Asian outflow plume. Such a split of a single outflow plume caused by a blocking



high pressure system over the Pacific is also described by Heald et al. (2003). Though in that particular case study, the northern branch of the plume reached the western US while the second branch headed southward towards the tropical western Pacific and no transport towards the Arctic was observed. Liang et al. (2005) and Reidmiller et al. (2010) also find that enhanced transpacific transport is characterised by the combined effects of a strong Pacific High and a strong low over Alaska.

Even though many studies investigate the effect of long-range transport, they are mostly based on case studies like discussed above, use a composite approach or focus on meteorological conditions leading to uplift of polluted air masses into the free troposphere. Our study presents a detailed analysis of the spatial and temporal distribution of elevated CO level as a pollution tracer in the mid and upper troposphere over the Pacific using 20 years of MOPITT data. We create a climatology of severe pollution episodes and use trajectory calculations to link each particular pollution event detected in MOPITT satellite data

with a distinct source region. A second objective is the investigation of different uplift regions and uplift types, in particular wcb-related upward transport. We analyse each trajectory linking a pollution event detected by MOPITT with a CO source region individually. We create a seasonal statistic about different transport pathways and uplift types depending on the location of elevated upper tropospheric CO, its source region, the uplift region and uplift type of polluted air masses.

## 2 Data and Model description

### 2.1 MOPITT

To detect pollution outflow events from the Asian continent, we use thermal infrared level 3 data from the version 8 product of CO measurements derived from the MOPITT instrument (Deeter et al., 2019). Level 3 products are available as daily mean values on a 1x1° global grid. The Terra satellite carrying the MOPITT instrument is flying in a sun synchronous polar orbit at an altitude of 705km. MOPITT splits the earth in pixels of a size of 22km$^2$. By using a cross-tracking scanning method it

sees the earth in a swath of about 640 km consisting of 29 pixels. A global coverage of the measurements would be reached after three days. Pixels with a high cloud content are filtered out. As MOPITT uses gas correlation spectroscopy of the thermal infrared radiation emitted from the earths surface, it can retrieve vertical profiles for almost two independent layers of CO. Data products are available on 10 level with a vertical resolution of 100hPa (surface, 900hPa - 100hPa). A retrieval algorithm is applied to the MOPITT data which is based on optimal estimation using a priori information to obtain additional constrains

(Deeter et al., 2015).

### 2.1.1 Method of data analysis

To be less dependant on the absolute number concentrations and related uncertainties of CO data derived from the MOPITT instrument (e.g. due to potential undetected slow drifts of the data over the 20 years of available data (Yoon et al., 2013)) and in order to capture only severe pollution outflow events from Asia, we select daily CO data in the following way: At each level,

we only take into account those grid points with a mixing ratio belonging to the globally highest 2% mixing ratios (see example





in Fig. 1, bottom row). We refer to those data points as CO-maxima. Regions with at least three neighbouring maximum-grid points are defined as a CO-maximum cluster and are included in or analysis.

## 2.2 MPTRAC

In order to link the regions of elevated CO derived from MOPITT with a particular source region, we use the trajectory model

MPTRAC (Hoffmann et al., 2016) which was recently developed at the Jülich Supercomputing Centre. MPTRAC is a massive-parallel lagrangian particle dispersion model allowing a computationally efficient calculation of transport simulations in the troposphere and stratosphere.

We calculate backward trajectories starting in a square region covering each single CO-maximum cluster detected by MOPITT. To account for the vertical extent of the averaging kernel and uncertainties in the reanalysis data set driving MPTRAC, trajec-

tories are initiated at 400hPa with a random variation of this altitude between 386hpa and 424hpa. The number of trajectories for each square covering a CO-maximum cluster is defined by ($\Delta$lon*$\Delta$lat)*100. Trajectories are started four times a day (00 UTC, 06 UTC, 12 UTC, 18 UTC) if a CO-maximum was observed. The simulation time is 16 days. Meteorological input data for all trajectory calculations are taken form the ERA-INTERIM data set (Dee and co authors, 2011). Output is also written every six hours in order to compare our MPTRAC simulations with the ERA-INTERIM data set. Trajectories are included in

our statistics if they descend below 850hpa, altitude above ground is less than 1.5km and if trajectories cross emission regions with a CO flux of at least $0.1 \times 10^{-9}$ kg(CO)/m$^2$s. This value roughly corresponds to an average CO emission flux in industrialised regions excluding biomass burning emissions. As a reference for CO emissions we use the IPCC AR5/RCP85 emission inventory (Lamarque et al., 2010).

## 2.3 Transport characteristics

One of the questions we address in this study is the potential importance of vertical transport of pollution by extratrocpical cyclones and particularly warm conveyor belts as part of long-range transport events. Closely following Madonna et al. (2014), we define a criterion for wcb related upward transport: trajectories must be located within a two-dimensional surface cyclone field for at least one 6-hourly time step during the ascend phase. i.e. trajectories must cross the cyclone within the time between two time steps prior or after the lift above the 800hPa level (Fig. 8).

This condition does not only ensure that trajectories rise in the vicinity of a cyclone but it also excludes accidental consideration of convectively uplifted air masses (Madonna et al., 2014). We require uplift from below 800hPa to 400hPa (the altitude we mainly analyse, see section 3.1) but we give no time limit for the uplift. Our analysis reveals however, that by far the majority of all trajectories is lifted to 400hPa within 48 hours which is the time frame for wcb-type uplift used in the study by Madonna et al. (2014).

To define the position and size of a cyclone we follow the method of Wernli and Schwierz (2006) and the modifications described by Madonna et al. (2014): Based on ERA-INTERIM reanalysis data a cyclone centre is defined as a local minimum in the sea level pressure. Starting at the centre of a cyclone, closed isobars are calculated in intervals of 0.5 hPa. To define the horizontal extension of a cyclone, this procedure is repeated until no closed isobars can be found. Like described by





Madonna et al. (2014) our algorithm also allows the merging of two cyclones which are very close to each other. The algorithm
calculating the horizontal extension of a cyclone tends to rather underestimate than overestimate the size of a cyclone.

In addition to wcb type uplift, we determine trajectories being lifted along a frontal zone. The classification of trajectories
belonging to this category follows the procedure described above but trajectories must cross a frontal zone during their ascent.
The definition of a front follows in general one of the approaches discussed by Schemm et al. (2017): a frontal zone is identified
by enhanced $\theta_e$ gradients (at least 4K/100km). We slightly modified this criterion by requiring a gradient of at least 3K in one
grid box (0.75°) in the ERA-INTERIM data set. As discussed by Thomas and Schultz (2018) numerous definitions to detect
a front in a gridded data set exist, all of them having their advantages and disadvantages. As the definition of the front has a
rather high uncertainty and as the ERA-INTERIM data set has a rather coarse horizontal resolution of 0.75° it is likely that we
miss frontal systems in our analysis. Therefore several trajectories do not fulfil one of the two above described criteria and are
classified as 'rest'. Also trajectories which are orographically lifted belong to this group. Due to the coarse vertical resolution
of the ERA-INTERIM data set which also underlays the MPTRAC trajectory calculations, it is not possible to separate those
trajectories accurately.

The ERA-INTERIM based MPTRAC simulations do not explicitly represent deep convection since trajectory calculations
are only driven by large scale wind fields. Therefore we cannot further distinguish between different uplift types occurring
in the vicinity of a frontal zone. Due to the long simulation period of 19 years it is also not possible to analyse each uplift
event individually as done by Liang et al. (2005). However, Lawrence and Salzmann (2008) point out that trajectories should
represent the net vertical and long-range transport reasonably, since much of the convective upwelling in our region of interest
is connected to large scale circulations. As Lawrence and Salzmann (2008) and Lawrence and Lelieveld (2010) discuss in
detail, it can be assumed that the basic regional lofting will be present in lagrangian trajectory simulations using input from
global circulation models. Though it can be expected that the mean rate of vertical transport is underestimated.

## 145  3  Results

### 3.1  CO enhancements observed by MOPITT

In order to investigate the impact of the continental Asian outflow on mid- and upper tropospheric pollution level, we analyse
CO data from MOPITT on different pressure level. As mentioned above, we only include the highest 2% of all daily available
data points for each level in our analysis. We refer to them as CO-maxima events. As we are primarily interested in upper
tropospheric pollution episodes, we focus our analysis on the 400hPa data from MOPITT. We also detect regularly CO-maxima
events over the NH-Pacific at 500hPa and 300hPa. Though during winter time, no clear CO-maxima signal is visible over the
northern Pacific at 300hPa. North of about 45°, the 300hPa level is frequently located above the tropopause during winter
(Wilcox et al., 2012) where CO mixing ratios decrease quickly.

Figure 1 (upper and centre row) shows seasonal mean CO mixing ratios from 2000-2019 at 400hPa, including all available
level 3-MOPITT data points. Throughout the year, CO mixing ratios over the NH-Pacific belong to the highest mixing ratios
observed globally, especially during winter, spring and summer. During autumn mixing ratios are high over typical biomass

**Figure 1.** Seasonal mean CO mixing ratios [ppb] from MOPITT at 400hPa including all available data points (2000-2019). Plots in the bottom row show daily CO mixing ratios from MOPITT for two selected days. Black crosses (bottom row) mark the 2% highest CO values of the given day, which are selected. Black squares mark the region in which trajectories are initiated.

burning regions (South America, southern Africa) and their outflow regions over the southern hemispheric Pacific and Indian Ocean.

To identify individual synoptic scale pollution events and analyse them individually, we filter MOPITT data as described above. By only using 2% of all daily available data points we include roughly 80-130 grid points per day in our analysis (black crosses, Fig. 1, bottom row). As we select those grid points, independently where they are, it is hypothetically possible that on days with strong and widespread biomass burning at any region outside the studying region, the pollution signal over the NH-Pacific is weak and we underestimate the number of severe pollution events. Though, due to the incomplete global coverage of the satellite data, MOPITT only sees a fraction of the potential biomass burning area (Fig. 1, bottom row). The situation is only different during autumn when CO mixing ratios over southern Africa and America are extraordinary high and the biomass burning CO-signal is visible broadly over the southern hemispheric Atlantic and Indian Ocean (Fig. 1, SON). Indeed, we detect much less pollution events over the NH-Pacific during this time of the year (Fig. 3 SON) but, CO mixing ratios there are much





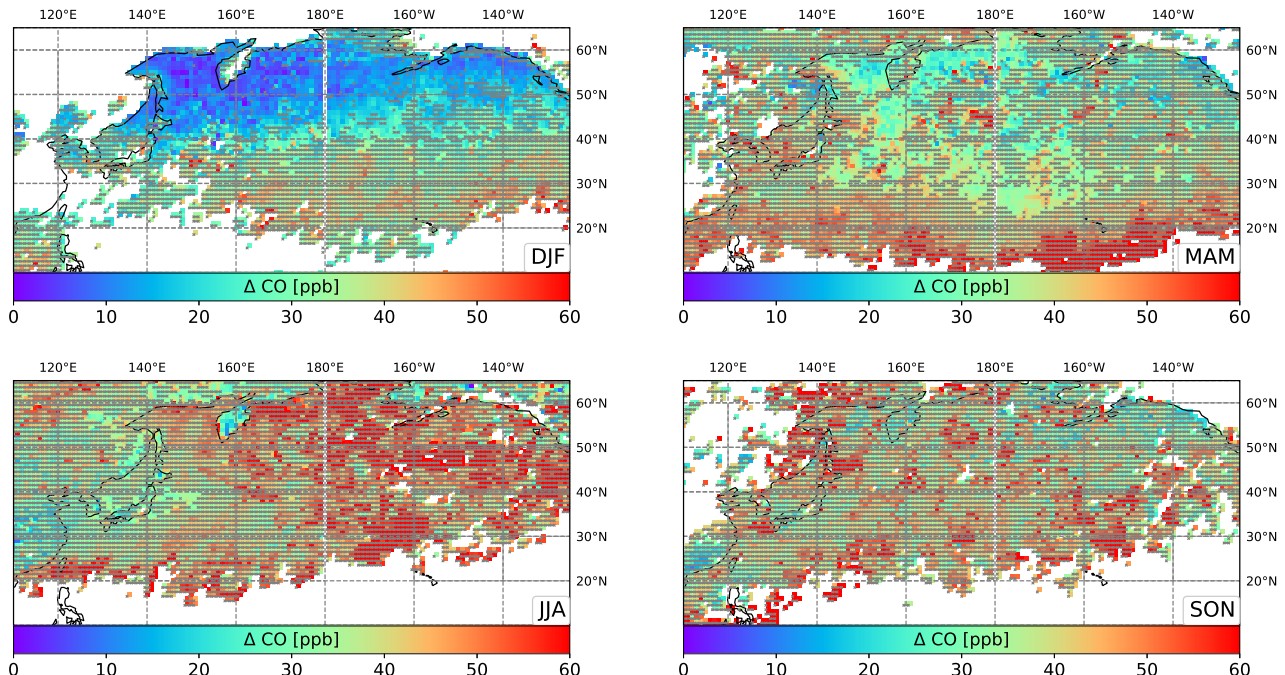

**Figure 2.** Difference (in [ppb]) between CO seasonal mean mixing ratios considering only CO-maxima events and total seasonal mean CO mixing ratios including all valid data points (2000-2019). Grid points which are marked by grey crosses have a difference with a strong statistical significance (regions where the pooled standard deviation is within a 99% confidence interval).

lower during autumn than during the rest of the year.

In general, CO-maxima events represent periods of elevated CO mixing ratios compared to background level. The difference

between mean CO mixing ratios considering only CO-maxima events and mean CO mixing ratios considering all data is rather high over almost the entire NH-Pacific during all seasons (Fig. 2). This indicates, that by our selection of data points we really capture periods with extraordinary high pollution level in the upper troposphere. An exception is clearly visible over the northern Pacific during winter (Fig. 2, DJF) where the difference in CO mixing ratios is rather small and of no statistical significance. CO maxima events occur regularly in this region, though not significantly more often than over the central NH-

Pacific where the difference in CO mixing ratios between both data sets is much larger. Thus we can assume that CO level prevailing while the grid points were selected as a maximum, represent rather the climatological mean than a single extreme event. This conclusion can also be drawn for the central NH-Pacific during spring (Fig. 2, MAM). Regions with a difference in CO mixing ratios of low statistical significance are also found during summer east of Japan (Fig. 2, JJA) where many maxima events occur. Though, this region is much closer to potential CO sources than the northern/central Pacific. Thus it is less

surprising that pollution outflow events strongly impact the overall climatological mean at this site.

High CO level occur throughout the year over the NH-Pacific with autumn showing an exception with much less events than during the rest of the year (Fig. 3). A seasonal shift of the regions where most CO maxima occur at 400hPa is clearly visible





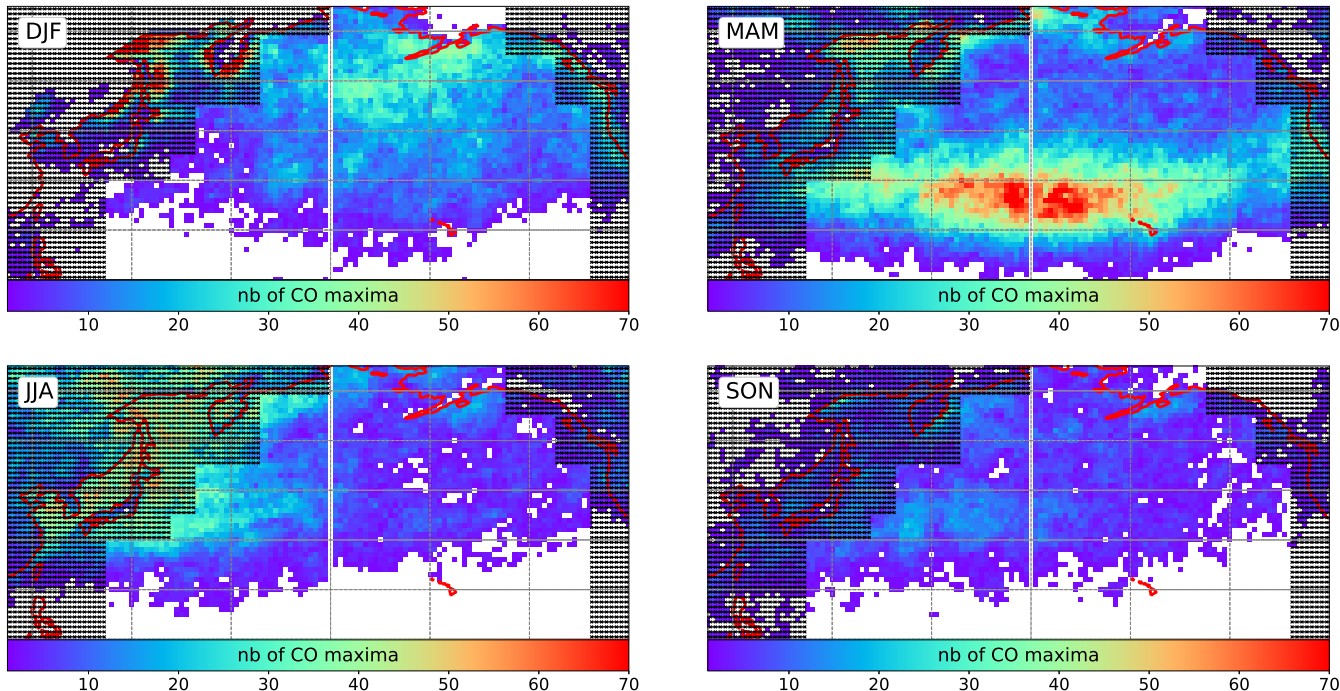

**Figure 3.** Number of CO-maxima events at a particular grid point for each season (winter (DJF), spring (MAM), summer (JJA) and autumn (SON)), summed up over the years 2000-2019. Only CO-maxima in the area which is not grey-shaded is included in the trend- and trajectory analysis.

(also at 500hPa and 300hPa). In winter time (Fig. 3, DJF), CO-maxima occur more often over the north-eastern Pacific while they are found over the central NH-Pacific during spring (Fig. 3, MAM). The total number of CO-maxima events is much

higher during spring than in other seasons. This agrees with other studies showing enhanced pollution export from Asia during spring (e.g. Luan and Jaeglé, 2013; Holzer et al., 2005; Zhang et al., 2008; Yu et al., 2008; Liang et al., 2004). During summer time (JJA) the passage of midlatitude cyclones is generally less frequent than during winter and spring while convection and the Asian summer monsoon gain in importance as transport pathways. The detected CO maxima are then found much closer to the continent over the western NH-Pacific, Siberia and Kamchatka.

The total number of CO-maxima events at a particular grid point over the NH-Pacific does not seem to be exceptionally high (Fig. 3). Many grid points are only selected 10-15 times as a CO-maximum point within one season in 20 years. On the example day shown (03.01.2001) we detect three CO-maxima cluster over the NH-Pacific (marked by black squares) consisting of 12, 18 and 12 grid points (Fig. 1). The small number of total counts of each single grid point is (i) the consequence of the incomplete spatial coverage of MOPITT (Fig. 1, bottom row), (ii) the rather high spatial variability of the position of the CO-

maximum cluster and (iii) the rather small number of grid points which we select daily. The number of valid data points over the northern Pacific is smaller than over other parts of the world (mainly due to clouds, Fig. 1, bottom row). Nevertheless, we detect CO-maxima over the NH-Pacific during ∼ 70% of all days during winter and summer and ∼ 80% during spring (Fig. 4 right

**Figure 4.** Plots show time series of the season mean number of daily CO-maxima grid points, normalised with the total number of valid MOPITT data points (left column). Circles are coloured with the ENSO index (taken from https://www.cpc.ncep.noaa.gov/products). Vertical lines show the standard deviation of the daily number of CO-maxima grid points with respect to the season mean. Further shown is the season mean number of CO-maxima cluster (centre) and the percentage of the number of CO-maxima days within one season (right). Excluded from the statistics are CO-maxima over the continent and coastal areas close to major CO source regions (grey area marked in Fig. 3 not included). For the trend calculation, MOPITT data until 03/2020 are included but periods with large data gaps (2 out of 3 months missing data) are excluded.


column). The number of grid points which are selected as a CO-maximum each day varies however, strongly (visible in the high standard deviation, Fig. 4 left column). During winter, the mean number of grid points which we select as CO-maxima, stays in

the same order of magnitude throughout the 20 years of available MOPITT data. Though during spring it increases from 2000 until 2020 and decreases during summer (Fig. 4 left column). At the same time, the total number of valid MOPITT grid points increases/decreases. To account for this effect, we show the seasonal fraction of CO-maxima grid points with respect to the total number of valid MOPITT grid points over the NH-Pacific (Fig. 4 left column). Still, the increasing/decreasing trend is visible. Both trends (MAM, JJA) are mathematically of statistical significance. Though the results need to be carefully interpreted.

At first, the standard deviation of the mean number of CO-maxima grid points found in one season is very high. Secondly, we cannot directly conclude, that the spatial coverage of severe pollution episodes increased, though it is likely. An increase of the number of CO-maxima grid points over the Pacific during spring could hypothetically be also the result of e.g.: (i) weaker biomass burning in other regions of the world, (ii) a faster dilution of biomass burning plumes, (iii) less clouds over the Pacific or more clouds over other polluted areas or (iv) instrumentational reasons. Since precipitation patterns over the Pacific

and the position of the jet stream can be altered during El Niño episodes (Breeden et al., 2021), and thus transport pathways of pollution plumes, we considered a correlation between warm phases of ENSO and severe pollution episodes. Years with an extraordinary large spatial extension of the upper tropospheric pollution plumes (e.g. 2004, 2009, 2016, 2019) are indeed related with El Niño periods (Fig. 4 left column). This conclusion can however, only be drawn for spring and is in agreement with the work by Breeden et al. (2021) who find an increase in storm track activity during La Niña years compared to neutral

or El Niño years.

The number of days on which we detect CO-maxima does not show a corresponding positive trend during spring (Fig. 4 right column, MAM). The percentage of CO-maxima days during spring is even lower between 2010-2019 compared to the years 2003-2009. During winter, summer and autumn the number of CO-maxima days shows a decreasing trend corresponding to the decreasing trend in the number of CO-maxima grid points. Though the linearity of the the data set is rather weak

(Fig. 4 right column, DJF, JJA, SON) and therefore the trend. We also calculated the season mean number of CO-maxima cluster (Fig. 4 centre column) which however, does not change significantly during spring leading to the conclusion that the spatial coverage of individual pollution cluster increases in spring during the 20 years we analysed.

## 4   Specification of CO emission regions

### 4.1   Case study

As described in section 2.2, we use lagrangian trajectory calculations to analyse the source regions of upper tropospheric CO detected over the Pacific. To better illustrate the trajectory analysis, we will discuss in more detail two particular days (03.01.2001, 21.01.2001) with CO-maxima being observed over the remote Pacific by MOPITT (Fig. 5). On both days a CO-maximum cluster was found near the Aleutian peninsula (Fig. 5 b,d) and over the central NH-Pacific (Fig. 5 a,c). On 03.01, a

third cluster was found over the gulf of Alaska (Fig. 5 e).





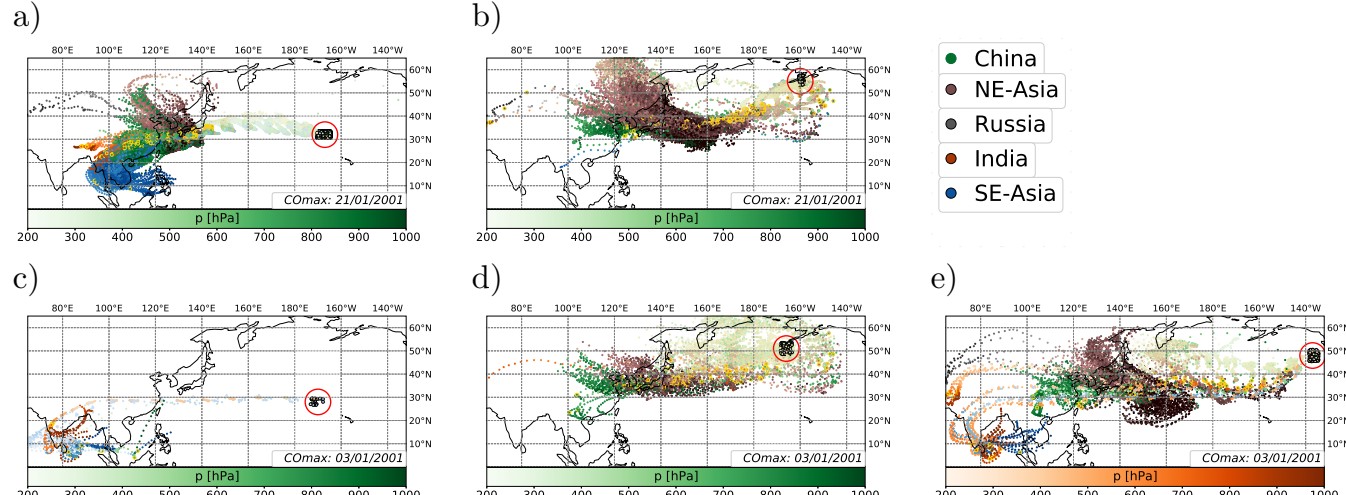

**Figure 5.** Plots show trajectories starting at a CO-maxima cluster on 03.01.2001 (lower row) and 21.01.2001 (upper row). Trajectories with the same source region, have the same colour (see legend, e.g. trajectories from China are green, those from SE-Asia blue). The lighter the colour of the circles marking the position of the trajectory, the lower the pressure level. Yellow circles mark the time of uplift, black circles (over the Pacific) mark the start position of the trajectories, i.e. the region of the CO-maxima cluster.

Trajectory simulations link all three CO-maxima cluster north of 30°N (Fig. 5 ,b,d,e, Fig. 6 ,a,d) with NE-Asia (dark red trajectories) and NE-China (green trajectories) as source regions while the cluster over the central NH-Pacific (Fig. 5 a,c) have a strong contribution from air masses coming from (SE-)China (green trajectories) in addition to air masses coming from SE-Asia (blue trajectories) and India (orange trajectories). Even though both cluster over the central NH-Pacific are found in very similar regions (∼ 170°E, 30°N, Fig. 5 a,c), trajectory simulations indicate significantly different source regions: On 21.01 (Fig. 5 a, Fig. 6 a), 65% of all trajectories come from China, 15% from NE-Asia and SE-Asia and a small fraction of trajectories come from India (4%). While the CO maximum observed on 03.01 (Fig. 6 d) is mainly fed with CO from SE-Asia (46% of all trajectories) and India (50%) while China seems to play a minor role as source region (3%) on that particular day. Most of the trajectories reaching the CO maximum over the Aleutian peninsula on 21 January, are lifted into the free troposphere in the vicinity of an eastward moving cyclone (66% of all trajectories, Fig.6 c). Fig.7 shows exemplarily the position of the cyclone at 18:00 UTC, 14.01.2001 (a) and 00:00 UTC on 15.01.2001 (b). Once being lifted into the free troposphere, air masses are transported north-eastward, circulate in the Aleutian low, and finally move further north towards the Arctic or eastward towards the American west coast (not shown). Thus long range transport of polluted air masses to the northern CO-cluster on 21 January is typical of uplift of air masses in the warm conveyor belt of a midlatitude cyclone over the Pacific with subsequent eastward motion in the free troposphere (Madonna et al., 2014). In contrast, air masses reaching the second cluster on 21 January over the central NH-Pacific (Fig.5 a) are mainly lifted into the free troposphere over the continent (Fig. 6 b,c) along a front.

Air masses reaching the CO-maximum cluster over the Aleutian peninsula on 03 January do not show such a uniform transport





pattern as seen on 21 January. Even though a large fraction of trajectories experience wcb-type uplift (43% of all trajectories,

Fig.6 d), the uplift time and uplift position has a stronger temporal and spatial variability compared to 21 January. Air masses need on average 3.3 days to reach the CO-maxima cluster after rise into the free troposphere (4.8 days on 21.01.). Parts of the air masses are lifted faster to higher altitudes ($\sim$ 400hPa 6-12 hours after uplift, Fig. 7 e, trajectories at the edge of the cyclone at $\sim$ 165°E), compared to 21 January, but transport patterns in the free troposphere are rather diffuse (Fig.7 d,e). On both case study days, a strong Aleutian low has built up. Though it is located over the gulf of Alaska on 03 January (Fig.7 f) and over the

Aleutian peninsula on 21 January (Fig.7 c).

The pollution plume over the gulf of Alaska (Fig. 5 e) is half composed of air masses which were lifted near CO emission regions (NE-China, NE-Asia) and eastward transport took place in the free troposphere while the other half crossed almost the entire Pacific in the lower most (marine) troposphere. Those air masses are finally lifted into the free troposphere close to the observed CO-maximum cluster in the vicinity of a cyclone. As a consequence it can be assumed that both plume composites

experience significantly different chemical processing and mixing during transport.

## 4.2  Statistics of long-term observational data set (2000-2018)

We extended the above presented statistics to all CO-maxima cluster detected in our simulation period (03/2000-12/2018) and to all trajectories being calculated for each single CO-maximum cluster. As described in section 2.2, backward trajectories are

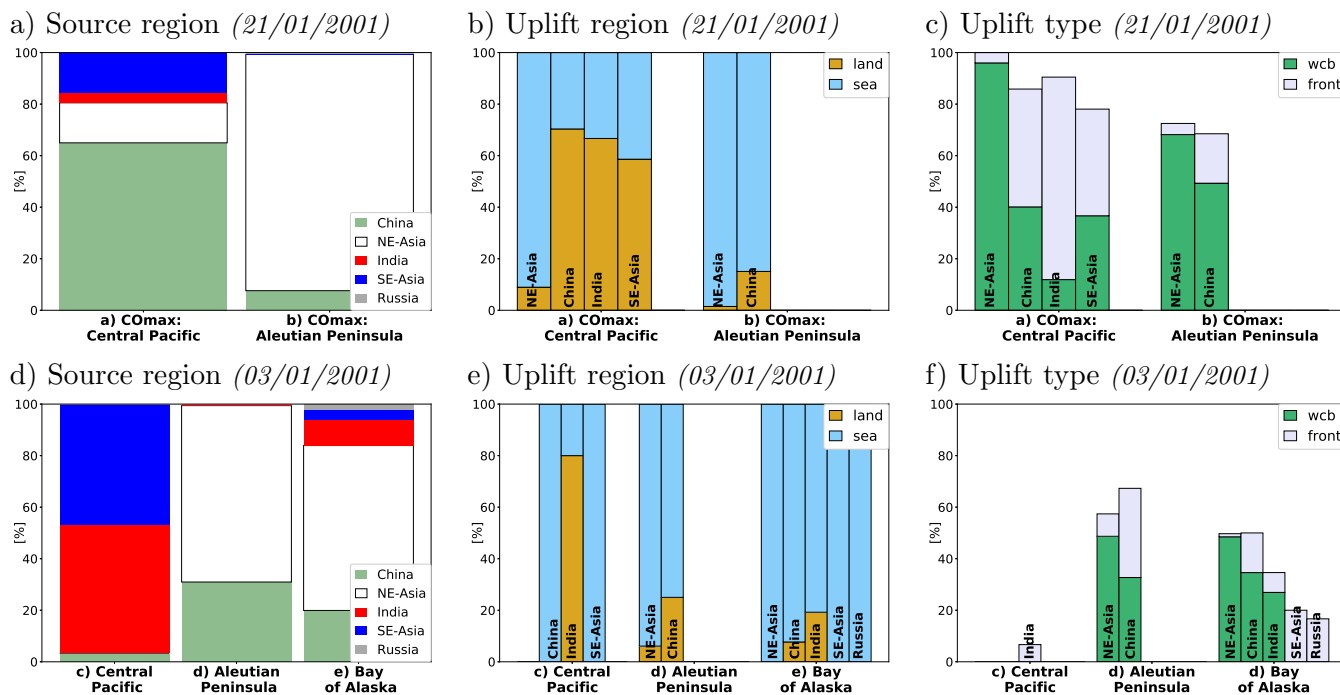

**Figure 6.** Plots show statistics for both case study days. The x-axis label refers to the different CO-maxima cluster shown if Fig. 5. Source regions are shown in Fig. 8 a.





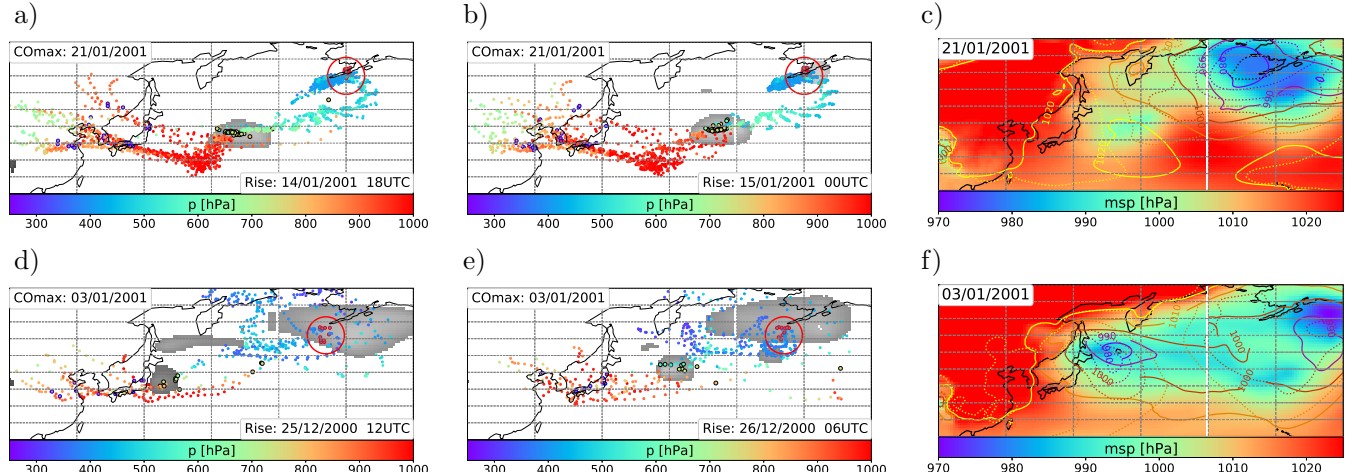

**Figure 7.** Plots (a-b),(d-e) show trajectories reaching a common CO-maxima cluster and having a common time of rise (above 800hPa). Black circles mark the trajectory position at the time of rise, blue circles mark the position of descend below 800hPa above a CO emission region. Grey shaded areas mark the position of a low pressure system in the ERA-interim data set. Plots c) and f) show a daily average of the mean sea level pressure [hpa] taken from ERA-interim on 21.01.2001 (c) and 03.01.2001 (f). Solid contour lines show the daily mean msp one day prior the case study day (20.01.2001 (c), 02.01.2001 (f)) and dotted contour lines two days prior the case study day (19.01.2001 (c), 01.01.2001 (f)).

assigned to a certain source region, if they descend below 850hpa, are less than 1.5 km above ground and CO emissions exceed
a certain threshold. Throughout the year, trajectories reveal China as the dominant source region (Fig. 8 d) of upper tropospheric CO. The contribution of other regions shows a stronger seasonality: during winter and summer almost the same percentage of trajectories come from NE-Asia (33.7% DJF, 27.9% JJA) as from China (36.8% DJF, 29% JJA, Fig. 8 d). During spring however, India becomes an important upper tropospheric CO source region (32% source contribution). During summer the total percentage of trajectories coming from India is smaller than during spring. Though, we find that the individual contribution of
India to a single pollution event is stronger during summer than during spring.

Considering only CO-maxima over the NE-Pacific (north of 45 °N, Fig. 8 e), China and NE-Asia are the dominant CO source regions (statistics are similar as for the entire Pacific). Though, only considering CO-maxima over the southern NH-Pacific (between 0° and 30°N), we see a much stronger source contribution from SE-Asia (Fig. 8 f), especially during winter (50%) and autumn (45%) and respectively India during spring (38.4%) and summer (40.7%).

The yellow area marked in Fig. 8 a) covering Indonesia, has a source region contribution of less than 1% and is therefore (almost) not visible in the bar charts in Fig. 8.

The two case studies indicates, that pollution plumes reaching a CO-maximum over the NE-Pacific during winter coming from NE-Asia, are often lifted into the free troposphere in the warm conveyor belt of a mid-latitude cyclone over the Pacific. Statistics including all 19 years of trajectory calculations lead to the same result: More than 90% of all trajectories from NE-
Asia rise over sea during winter (almost 80% during spring) (Fig. 8 b, DJF) and 26% of these trajectories rise by wcb-type





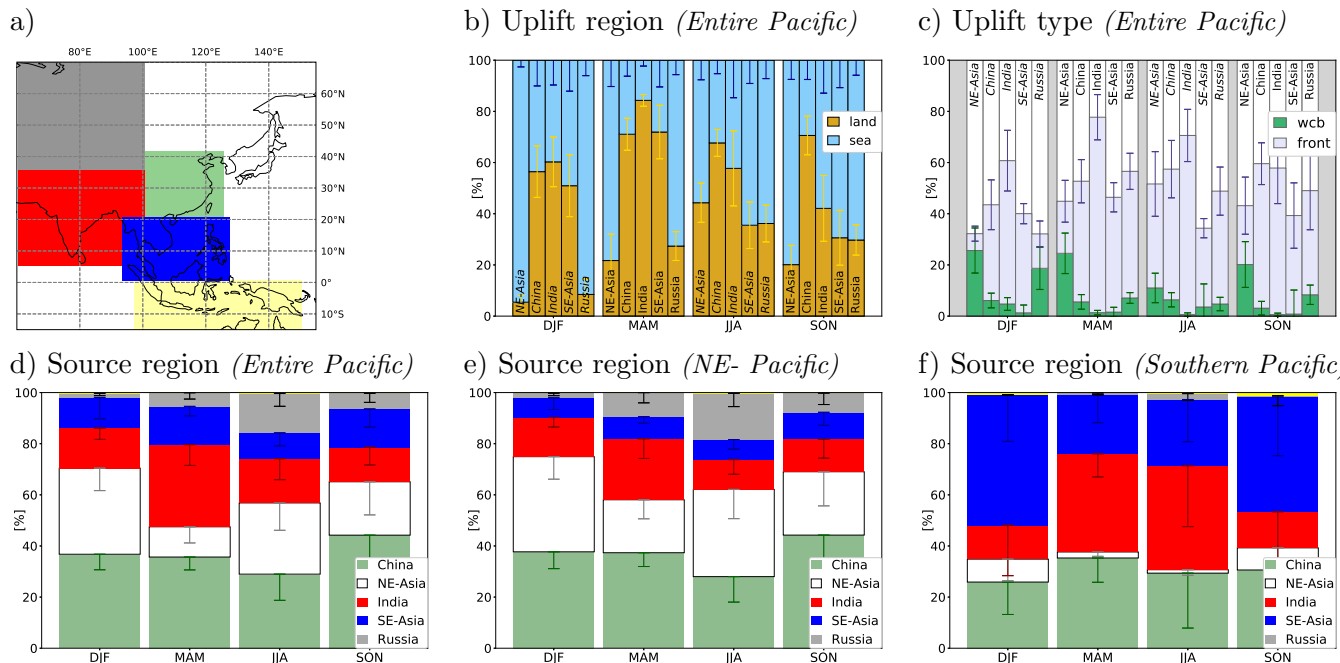

**Figure 8.** Source regions between which we distinguish are shown in (a). All other plots show seasonal statistics (in [%]) of the trajectory calculations: Fig.8(b) shows the percentage of trajectories rising over land or sea in dependency on the source region. The uplift type is shown in (c). Plots (d-f) show the percentage of trajectories coming from a particular source region (i) including all CO-maxima over the entire Pacific (d), (ii) including only CO-maxima north of 45 °N (e) and (iii) including CO-maxima only between 0° and 30°N (f). Vertical lines show the standard deviation (only plotted in the negative direction in plots (d-f)).

uplift (Fig. 8 c, DJF). Only taking those trajectories as a measure for which we can determine the type of rise, more than 90% of those are classified as wcb-type.

The fraction of wcb-related upward transport is much smaller for all other source regions throughout the year. This finding is in agreement with the detailed study by Madonna et al. (2014) who show that by far the largest fraction of trajectories experiencing wcb-type uplift over the Pacific come from NE-Asia especially during winter. This can be explained with the position of a semi permanent low pressure system over Japan during winter time (Liang et al., 2005; Madonna et al., 2014). For trajectories from all other source regions, uplift along a front line is the dominant uplift process. The uplift region shows a strong seasonality for all source regions. Only trajectories coming from China are predominantly lifted over land throughout the year (more than 60% with a rather small standard deviation, Fig. 8 b).

## 5 Conclusions

We analysed 20 years of CO data from the MOPITT satellite instrument with a particular focus on long-range transport of Asian pollution to the extratropical upper troposphere over the remote NH-Pacific. Our study indicates that those long-range transport





events occur regularly and with a high frequency in particular during winter and spring (on average more than one long-range transport event per week during spring and roughly one event per week during winter and summer). Additionally, the spatial

coverage of upper tropospheric CO-plumes seams to increase during spring within the 20 years of analysed data. We find that regions where most CO-maxima occur show a strong seasonal cycle with most events being detected over the north-eastern Pacific during winter, over the central NH-Pacific during spring and over the western NH-Pacific during summer. This shift of pollution patterns indicates significantly different transport pathways depending on the season. In addition, the contribution of different CO emission regions to the observed upper tropospheric CO plumes change among the seasons. Though China is

the major CO source region throughout the year. In relation with uplift in the warm conveyor belt of a midlatitude cyclone, NE-Asia is the second most important source region for upper tropospheric CO observed over the north-eastern Pacific in particular during winter.

The statistical analysis of our trajectory simulations reveal a correlation between certain CO source regions, uplift types, uplift region and upper tropospheric pollution regions. Though a more detailed analysis of single case studies, reveal that the

transport time of air masses reaching one single CO-maximum can differ strongly. Even though we describe transport patterns for one of our case studies (21.01.2001, CO-max cluster over the Aleutian Peninsula) as rather uniform, the time of uplift of air masses in the vicinity of a midlatitude cyclone into the free troposphere differs by more than 24 hours for single trajectories, corresponding to hundreds of kilometres in distance. Therefore, it is almost impossible to create composites of similar events in order to define transport regimes in relation to the prevailing synoptic situation, source region and upper tropospheric pollution

location.

In general, the time between rise into the free troposphere and the arrival of air masses at the CO cluster is longer for trajectories coming from China (3.7 days DJF, 6.6 days JJA), compared to NE-Asia (2.8 days DJF, 5.7 days JJA) corresponding to different uplift regions (land/sea). The residence time in the free troposphere is longer in summer than in winter for all source regions even though CO-maxima are found much closer to the continent during summer compared to winter. It is not surprising that

trajectories coming from India have the longest residence time in the free troposphere before reaching the CO-maxima site (6.5 days DJF, 9 days JJA). As a consequence of different transport patterns we can assume that the chemical composition and processing of Asian pollution plumes differs significantly depending on the source region and transport pathway.

The observations of CO maxima at 400hPa confirms the crucial role of east Asian emissions for the pollution of the lower UTLS. Notably this also holds for non-monsoon seasons. Our analysis presents evidence for a surprisingly regular and highly

frequent occurrence of these long-range transport events. Taking CO as a general marker of pollution and given the regularity of the transport processes, our study highlights the global role of the region also for other chemical constituents in the upper troposphere.

*Code and data availability.* MOPITT data were obtained from https://www2.acom.ucar.edu/mopitt. ECMWF (ERA-Interim) data have been retrieved from the MARS server. The code from MPTRAC is available from https://github.com/slcs-jsc/mptrac.





*Author contributions.*  PH designed the study. LS performed MPTRAC simulations, processed and analysed the data (MPTRAC, MOPITT, ERA-INTERIM) with input from PH. LS wrote the paper with input from PH.

*Competing interests.*  The authors declare that they have no conflict of interest.

*Acknowledgements.*  This research has been supported by the Deutsche Forschungsgemeinschaft (grant no. HO 4225/13-1).
The study is a pre-study related to the PHILEAS mission (Probing high Latitude Export from the Asian summer monsoon, HO 4225/17-1)
within the HALO-SPP 1294.



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
