# Peer review of "Contribution of Asian emissions to upper tropospheric CO over the remote Pacific"

_Atmospheric Chemistry and Physics, 2021_

## Author Comment (AC1)

At first we want to thank the reviewer for the very helpful comments to improve the manuscript.

*A general remark by the authors:*
*We should have stated more clearly, that it was NOT our objective, to find ALL severe pollution events over the Pacific during the last 20 years.*

*We decided to chose a filter (among many possible filtering methods using either numerical models, satellite data or other observational data), which ensures that really all events which we select represent episodes of very high CO level in the upper troposphere.*
*Therefore, the statistics we present in section 3.1, refers solely to our specific selection of pollution events.*

*Overall we analyze 17232 individual pollution cluster. We consider this number as sufficient to create a statistics (in particular of the related trajectory analysis). The number of individual events which we analyze is much larger than in comparable studies from e.g. Luan and Jaeglé (2013), Liang et al., 2005, 2004, 2007.*

*The major objective of our work, was to identify pollution events using MOPITT data and link them to source regions and uplift mechanisms.*

*→ We rephrased the last paragraph of the introduction ('objectives') and the abstract to emphasize the motivation and objectives of our work.*

*→ We also rephrased section 3.1*

**Abstract: the latitude range of the study region should be specified in the abstract.**

**There are several instances where a sentence is started with "Though…" Without a following clause, these are not complete sentences and are confusing in terms of meaning. Either connect these to the previous sentence or use a different starting word for example:**

**L22, "Though these studies…" should be "However, these studies…"**

**L144: "Though it can be expected…" Do you mean "Nonetheless, it can be expected…"?**

**L179: "Though, this region…" I think you can make this one sentence with: "…events occur, even though this region…"**

**L39 and throughout – I would capitalize the WCB acronym – wcb looks a bit like web and could be confusing to the reader.**

**L107 "As a reference for CO emissions…" This is confusing wording – could imply that MPTRAC uses an emissions inventory, which I don't think is the case. Maybe say explicitly: "We use the IPCC AR5/RCP8.5 emission inventory (Lamarque et al., 2010) to determine emission regions above the threshold.**

**L124 "extension" should be "extent"**

**L173 "extraordinary high pollution" => "significant high pollution"**

**L175 sentence starting with "Thus we can assume.." is confusing. Please re-word**

**L180 "level" can be confused with vertical levels – maybe use "High CO events"**

*→ We rephrased these sections of the manuscript!*

***L90 By using the grid cells with the highest 2% mixing ratios, you still have a seasonal dependence on the contribution of the a priori to the denominator CO amount due to differences in MOPITT sensitivity over seasons, as well as the global contributions from seasonal SH biomass burning (as you acknowledge). The contributions of these effects should be determined, including the trends in biomass burning (e.g. Andela et al., 2017) and how they affect the selection of pollution outflow events should be quantified.***
***Reference:***

***Andela, D.C. Morton, L. Giglio, Y. Chen, G.R. van der Werf, P.S. Kasibhatla, R.S. DeFries, G.J. Collatz,S. Hantson, S. Kloster, D. Bachelet, M. Forrest, G.Lasslop, F. Li, S. Mangeon, J.R. Melton, C. Yue, J.T. Randerson, A human-driven decline in global burned area, Science, 356 (6345) (2017), pp. 1356-1362, 10.1126/science.aal4108***

1) Please refer to our general remark at the beginning of our reply.

2) It would be rather difficult to determine quantitatively the effect of biomass burning on our selection of pollution events as we cannot distinguish between CO sources by solely using MOPITT data. We can only assume that most of the CO which is observed over the southern hemisphere and over central Africa is emitted by fires.
Finally, it was not our objective to investigate the impact of biomass burning on observed CO level.

3) The choice of the 2%-filter is originally based on an analysis of the frequency distribution of measured CO mixing ratios (we will add a figure showing this frequency distribution to the manuscript). These have a gaussian like distribution at 400 hPa. Literally spoken, we 'cut off' the right tail of the distribution and analyze only this part of the MOPITT dataset.

We considered it as surprising, that many of these 2% grid points are found very regularly over the remote Pacific – also during winter and spring when biomass burning is known to lead to high CO emissions over Central Africa and South America.
This gives the motivation for this manuscript: Quantification of the source regions and long-range transport pathways underlying these high levels of CO in the upper troposphere far away from strong CO sources.

It is a strength of our 2%-method, that CO lifetime effects or seasonalities would not strongly affect the relative contributions of our daily frequency distribution. The 2% tail of the frequency distribution is affected in the same way by ambient conditions as the remaining 98%.

To compose a robust statistics about the CO source regions and CO transport mechanisms (our major objective!), it is necessary to include a large number of pollution events in the analysis which is given by the 2%-filtering method. The average number of pollution events for which we calculated trajectories is: 195 for DJF per year, 330 for MAM per year, 239 for JJA per year, 98 for SON per year.

As the total number of events in our trajectory analysis is rather large (17232 individual pollution cluster) and events are distributed over 18 years, our source region and uplift statistics is not impacted by the fact that the number of events (selected with the 2% criterion) varies slightly during time or that we miss some events (e.g. due to clouds over pollution cluster).
The increase of the number of selected grid points over the Pacific during NH-spring is however, striking. Therefore, we discuss hypothetical reasons for this increase (one is a change in biomass

burning).  A more detailed analysis of the MOPITT data was however, beyond the scope of this manuscript and as mentioned above, would not change our main results (the trajectory analysis).

As we mentioned above, we rephrased section 3.1 to point out that the statistics (especially fig. 4) solely refer to our individual selection of pollution events (which however, are not chosen randomly but follow a mathematical criterion) and have to be considered as additional information to interpret the trajectory data.

At the end of this reply we add a figure showing fig. 4 of the manuscript for Central Africa, South Africa and South America. This gives an indication about the seasonal and regional distribution of the 2% grid points.

**L101 "number of trajectories is defined by delLat x delLon x 100" What is the range and typical value for this number?**

The size of the domain covering a cluster varies strongly. It ranges from rather small areas having a size of 2°x2° degrees up to areas having a size of 5°x6°. Larger cluster are detected rarely (roughly once per month).

The cluster shown in fig. 1 (lower most row) are therefore of average size. The squares covering the five cluster being discussed as case studies have a size of $(6°x3°)^a$, $(2°x4°)^b$, $(3°x6°)^c$, $(5°x5°)^d$ and $(4°x5°)^e$.

For these five cluster we calculated $1800^a$, $800^b$, $1800^c$, $2500^d$ and $2000^e$ trajectories at four different start times. Thus the final number of trajectories which we calculated for each cluster is: $7200^a$, $3200^b$, $7200^c$, $10000^d$ and $8000^e$.

We performed sensitivity simulations calculating e.g. twice the number of trajectories for each cluster. As the statistics regarding the source region contribution did not change significantly we decided to keep the number of trajectories as small as possible (but still sufficient!) for computational reasons.

**L105-106: What is the sensitivity of the results to these threshhold values?**

We have performed indeed sensitivity tests regarding these threshold values. The sensitivity of the results was very small regarding the CO emission flux as almost all CO source regions (apart from Siberia) have rather high CO emissions throughout the year.

The results (trajectory statistics) are also not very sensitive regarding the exact criterion for the descend into the boundary layer. The total number of trajectories following this criterion is very large and almost all of these trajectories reach the surface level (in the ERA-INTERIM data set used to drive MPTRAC) quickly after descend below 850 hPa within one source region. As we defined each source region as a rather large area, it happens only rarely that a trajectory crosses a 'source-region border' while being in the boundary layer within 24 hours after descend.

**Figure 2: Grey crosses are difficult to distinguish- maybe use these to indicate areas without statistical significance. Also, please state pressure level in the figure caption.**

We added "400 hpa" to the figure caption.

***L201 "At the same time, the total number of valid MOPITT grid points…" Does this analysis consider the number of MOPITT retrievals per grid cell? This could also be indicative of sampling changes over time due to clouds.***

Yes, indeed. We replaced fig. 3 (showing so far the total number of CO-maxima events which we detect) with a figure showing the number of CO-maxima events weighted with the number of MOPITT retrievals for each grid cell to exclude sampling changes.

***L263 – Sec. 4.2 How do these results agree or disagree quantitatively with previous results using different approaches? A broader discussion and/or table would be useful.***

→ *Conclusions: We extended and rephrased this section*

Comparable studies from e.g. Luan and Jaeglé (2013), Liang et al., 2005, 2004, 2007 use numerical models and/or satellite data to filter long range transport (LRT) events across the Pacific. They come to similar conclusions like us regarding the seasonal distribution of LRT events and the processes leading to uplift of pollution into the free troposphere. Some of these studies use however, a composite approach. This is a justified and well documented approach to investigate LRT events but, the analysis refers to the mean properties of the composite. As we analyze each pollution event individually, our results regarding transport pathways and transport times are more accurate. Furthermore, we analyze a much larger time period than comparable studies. This gives us the opportunity to assess the contribution of different CO source regions and their seasonal variability to CO observed over the Pacific.
In addition we are not aware that any other study so far has not pointed out that Asian emissions impact severely and regularly CO level in the upper troposphere (400hpa) far away from CO source regions.

***L307-310: I had a hard time following this logic and implications. Does this mean these cases are not included?***

All these cases are included in our statistics.

It would be desirable to find a clear correlation between the occurence of high levels of CO in the upper troposphere over the remote Pacific (e.g. over the Aleutian peninsula/north-eastern NH-Pacific during winter) and the prevailing synoptic situation.
This is very generally the case as we find common uplift mechanisms for trajectories having a common source region and a common start region (thus the region where CO maxima are found). Though, we find strong differences when comparing in detail individual pollution events with each other (regarding: location of CO maxima, location, strength and development of the Aleutian low, transport time from a CO-source region over the continent to the observation location of elevated CO over the Pacific, location of uplift from the boundary layer to the upper troposphere, position and strength of cyclones leading to uplift, residence time in the boundary layer/free troposphere during transport).
Therefore, we conclude that it is very difficult to further group (= create composites of) trajectories with similarities especially regarding a statistical correlation to meteorological parameter.

To better quantify the contribution of Asian emissions to north American pollution level however, it might be important to link the occurence of long range transport events and transport pathways in more detail with the prevailing synoptic situation over the Pacific.

**Central Africa**

[Figure]

**South Africa**

(A) Daily mean number of:
   COmax grid points

(B) Daily mean number of:
   COmax cluster

(C) Season mean number of:
   COmax days

[Figure]

**South America**

(A) Daily mean number of:
  COmax grid points

(B) Daily mean number of:
  COmax cluster

(C) Season mean number of:
  COmax days

---

## Author Comment (AC2)

At first, we want to thank the reviewer for his helpful suggestions to improve the manuscript.

*A general remark by the authors:*
*Apparently, the objectives of our work and the filtering method of the MOPITT data and its purpose did not become clear enough in our manuscript.*
*Therefore, we give an explanation at the beginning of this reply, hopefully clarifying misunderstandings.*

*(1)*
*We should have stated more clearly, that it was NOT our objective, to find ALL severe pollution events over the Pacific during the last 20 years.*

*In fact we wanted to identify long-term pollution in the UT on the basis of daily observations. For this purpose we used CO from MOPITT and assumed the highest 2% of the daily global observations to represent pollution.*
*We decided to chose a filter (among many possible filtering methods using either numerical models, satellite data or other observational data), which ensures that really all events which we select represent episodes of very high CO level in the upper troposphere (i.e. the globally highest 2% mixing ratios).*

*Therefore, the statistics we present in section 3.1, refers solely to our specific selection of pollution events.*
*Overall we analyze 17232 individual pollution cluster. We consider this number as sufficient to create a statistics. The number of individual events which we analyze is much larger than in comparable studies from e.g. Luan and Jaeglé (2013), Liang et al., 2005, 2004, 2007.*

*The objective of our work, was to identify pollution events in the upper most troposphere (other studies do not focus on this altitude range!) using MOPITT CO data and link them to (i) source regions and (ii) specific uplift mechanisms.*

*→ We rephrased the last paragraph of the introduction ('objectives') and the abstract to emphasize the motivation and objectives of our work.*

*→ We also rephrased section 3.1*

*(2) Idea behind the 2% filter:*

*The choice of the 2%-filter is originally based on an analysis of the frequency distribution of measured CO mixing ratios (we will add a figure showing this frequency distribution to the manuscript). These have a gaussian like distribution at 400 hPa. Literally spoken, we 'cut off' the right tail of the distribution and analyze only this part of the MOPITT data set.*

*We considered it as rather surprising that many of these 2% grid points are found very regularly over the remote Pacific – also during winter and spring when biomass burning is known to lead to high CO emissions over Central Africa and South America.*

*High levels of CO over the remote Pacific must be related to long range transport events as the locations where we detect these pollution cluster are far away from CO source regions.*

*(3) Selection of pollution events/long range transport (LRT) events:*

*As mentioned above, many selection criteria for LRT events exist.*
*The idea behind the approach of Luan and Jaeglé (2013) is very similar to our method: Based on modelled AOD data they create a frequency distribution of AOD values in a certain latitude range.*

*This distribution is log-normal (similar like the CO mixing ratio frequency distribution). They choose the top 20% days in the frequency distribution as LRT events.*

*To compose a robust statistics about the CO source regions and CO transport mechanisms (our major objective!), it is necessary to include a large number of pollution events in the analysis which is given by the 2%-filtering method. The average number of pollution events for which we calculated trajectories is: 195 for DJF per year, 330 for MAM per year, 239 for JJA per year, 98 for SON per year.*
*As the total number of events in our trajectory analysis is rather large (17232 individual pollution cluster) and events are distributed over 18 years, our trajectory statistics is not impacted by the fact that the number of events (selected with the 2% criterion) varies slightly during time or that we miss some events (e.g. due to clouds over pollution cluster).*

*→ As we mentioned above, we rephrased section 3.1 to point out that the statistics (especially fig. 4) solely refer to our individual selection of pollution events (which however, are not chosen randomly but follow a mathematical criterion) and have to be considered as additional information to interpret the trajectory data.*

**General Comments**

**The manuscript describes the use of satellite CO data for the 2000-2019 period to investigate trends over the Pacific region at 400 hPa. Trajectory calculations are used to determine the source regions of CO enhancements observed over the same area. MOPITT thermal infrared (TIR) L3 CO data from version 8 are used. This is an interesting topic and the MOPITT dataset is well suited to the goals stated.**

**Some key points need to be clarified in order to demonstrate if the data selection method is appropriate to reach the goals stated, though. According to the manuscript, "to capture only severe pollution outflow events from Asia" the highest 2 % MOPITT mixing ratio values (or CO-maxima) are selected. Regions with 3 or more neighbouring CO-maxima (a CO-maxima cluster) are then analyzed.**

**The adoption of the 2 % method seems to be based on "potential undetected slow drifts of the data over the 20 years of available data (Yoon et al., 2013)";**

This is a misunderstanding. The potential drift is only a minor criterion for our filter. Please see our explanation on top of the reply regarding the choice of the filtering method.

**the Yoon reference discusses version 5 of the MOPITT dataset. For completeness, the manuscript should consider other relevant studies. For example, Deeter et al. (2019) demonstrated that bias drift at 400 hPa (i.e., the pressure level analyzed and discussed in the manuscript) is 0.0 % / year for the MOPITT version 8 TIR dataset, which is the same version of the dataset analyzed in this manuscript. Since the premise of "potential undetected slow drifts" has not been sufficiently demonstrated, the use of the 2 % method needs to be better justified.**

→ We included the discussion of Deeter et al, 2019.

***Was the number of MOPITT L2 observations that went into each L3 data point taken into consideration? Some L3 data points are based on a few L2 observations, sometimes as few as 2. L3 data points based on a few L2 observations may not be representative.***

An analysis of L2 data was beyond the scope of this manuscript. We rely on the V8-MOPITT user guide and the related publication (Deeter et al., 2019) that L3 data which are publicly available (and has been used in other peer reviewed publications) have undergone a thorough quality control.

As we can link the locations above the Pacific having CO mixing ratios belonging to the globally highest 2% with CO source regions by our trajectory analysis (for individual events but also on a statistical basis) it appears to us that the observed signal in the MOPITT data is robust.

***It is not clear if the possible effects of missing data (due to clouds, calibration events, etc.) have been taken into consideration when calculating statistics and interpreting results. If they have, then the manuscript should clarify how. Missing data may follow seasonal/annual patterns and could affect the results and interpretations, if not properly accounted for.***

1) To account for missing data points, all quantities shown in Fig. 4 are weighted with the total number of valid data points (see caption of Fig. 4 and text p. 10, l.201/202).

2) Periods with data gaps are not considered for calculating the regression line. We forgot to mention this information in text and the label of Fig. 4 and thank the reviewer for this hint.

3) We exchanged Figure 3 showing the total number of CO maxima at a given grid point with a plot showing the total number of CO maxima grid points weighted with the total number of valid MOPITT data points (the same quantity plotted in Fig. 4, left column)

***The two lower panels in Fig. 1 seem to indicate that nighttime MOPITT CO values (half of the daily measurements) were not included in this analysis. This point should be clarified and, if true, justified.***

The two plots in the lower row of Fig. 1 show by purpose only daytime observations to demonstrate our filtering methods for which each overpath is considered separately.

***In general, global CO maps from MOPITT and other instruments show that pollution plumes emitted in China, other Asian regions, Siberia, etc. are transported across the Pacific and often reach North America and beyond. The manuscript mentions fires as the source of the CO; that would be the case for emissions from Siberia during the northern summer months and in some cases for emissions from SE Asia. However, most emissions from China are due to fossil fuel combustion and they occur during all seasons. This should be discussed in the manuscript.***

We do not state at any point in the manuscript, that we assign all CO to fire emissions. In Section 3.1, we discuss the (potential!) impact of biomass burning (in other regions on earth which are known to have rather low industrial emissions of CO but seasonally high CO emissions by biomass burning) on our selection of the CO maximum grid points.

Using solely MOPITT data, it is not possible to draw any further conclusions about specific CO sources (such as fires, traffic, and many others). Indeed our analysis shows (Fig. 7b) that the densely populated areas potentially contribute to upper tropospheric CO.

***The manuscript does not discuss the seasonal effects on CO lifetime and overall CO background values which would result in more frequent/persistent CO enhancements during the winter months.***

The seasonal variation of the CO lifetime is rather short compared to the total tropospheric lifetime of CO. The trajectory analysis reveals, that air masses need less than 10 days to be transported from CO source regions into the upper troposphere over the Pacific. Therefore, transport time is always shorter than the tropospheric CO lifetime.

In addition, it is a strengths of our 2%-method, that lifetime effects or seasonalities would not strongly affect the relative contributions of our daily frequency distribution. The 2% tail of the frequency distribution is affected in the same way by ambient conditions as the remaining 98%.

***The descriptions of results and their interpretations are hard to follow.***

Following the suggestions of the 2[nd] reviewer, we rephrased the 'conclusions' section.

**Specific Comments**

***An explicit definition of "elevated CO event", "severe pollution events" should be provided.***

***Do CO-maxima clusters represent elevated CO events/severe pollution events? Do CO values/statistics support the idea that CO-maxima cluster represent only severe pollution events as intended? CO-maxima clusters may or may not represent actual pollution events, since they are relative. It would be more appropriate to call those "relative daily CO maxima" or similar. Using an absolute CO threshold value would have resulted in the selection of absolute CO-maxima, most likely corresponding to severe pollution events only***.

1) Please see our general comment at the beginning of the reply.

2) "Elevated CO" is defined by the choice of the LRT events themselves (i.e. we only include grid points in our analysis with mixing ratios belonging to the globally highest 2% mixing ratios).

3) As CO mixing ratios differ regionally and seasonally strongly it would be rather difficult to chose a threshold mixing ratio to select pollution events. Therefore, we decided to chose a filter which is independent of absolute mixing ratios. As the difference between mean CO mixing ratios including all CO data and CO mixing ratios including the selected maxima is of statistical significance we can conclude, that we do select events of very high levels of CO.

***What are "neighbouring CO-maxima"? Do CO-maxima need to be adjacent to each other? Within some fixed distance?***
***Why is 3 the minimum number of neighbouring CO-maxima to form a cluster?***

On a quadratic grid, each grid point has 8 neighbouring grid points. We require, that at least 2 out of these 8 grid points are also selected as a CO-maximum (given then in total at least 3 grid points being neighbours). This is done to ensure that we only include larger regions of elevated CO in our trajectory analysis and not single grid points.

*It is unclear if the 2 % method does "capture only severe pollution outflow events from Asia". Elevated CO at 400 hPa (the only pressure level discussed in the manuscript) could potentially come from other regions, including Europe or even North America.*

Therefore, we performed the trajectory analysis to determine the source regions of the observed CO at 400hPa. Due to the long tropospheric lifetime of CO, emissions from other regions than Asia contribute to background CO level. Though, the CO-maxima cluster analyzed by us, are predominantly fed with CO from Asia. In addition to the source region, we can also identify distinct transport patterns.

We explain on p. 5, l. 147-153 why we analyze the 400hPa level.

We have restricted the analysis to 400hPa as we focus by purpose on the upper troposphere. We applied the filtering method also to other pressure level giving us also cluster of very high CO mixing ratios over the Pacific. Though, the 400hPa level is chosen as it is closest to the tropopause but reliably in the troposphere throughout the year. The 300hPa level is presumably (based on low CO mixing ratios) often stratospheric.

*Fig. 2. Blue (low) values in the upper half of the DJF map may indicate that the number of data points is very low. If the CO-maxima data points are basically the only data points available, then difference values will be close to 0. Season and geographical location are consistent with clouds resulting in a low number of data points.*

According to Fig.3a, we detected the highest number of events during DJF over the northern Pacific.  If the number of valid data points is small (but for sure not too small!) over the northern Pacific during winter but almost all valid data points belong to our selection of CO maxima grid points (which would lead to a small difference in CO in Fig. 2, DJF) our conclusion would not change: Based on considering all available data points, episodes of highly elevated CO mixing ratios determine the overall mean over the northern most Pacific during winter.

*Fig. 3, MAM panel. Could the region with high number of CO-maxima be a region with few clouds and, thus, more MOPITT observations which could result in more CO-maxima cases? VIIRS true color images show that this may be the case. ISCCP maps of seasonal mean cloud amount (%) support this point. If the statistical analysis did not account for the effect of clouds in the number of observations, then some of the manuscript results and conclusions may be invalid.*

→ Please see our general comment at the beginning of the reply.

By analysing the global distribution of the daily 2% highest CO mixing ratios we found these pollution cluster surprisingly often over the NH-Pacific. This gives the motivation of  this manuscript: We want to quantify the source regions and transport pathways explaining the finding of high levels of CO in the upper troposphere far away from strong CO sources.
In deed, we cannot detect pollution cluster in cloudy regions by using MOPITT CO data.

Therefore, Fig.3 gives the reader *solely* information about the locations of the CO cluster which are later on investigated in more detail by the trajectory analysis.
The reader can e.g. see that in DJF more trajectories are initiated over the northern NH-Pacific than in MAM. To account for seasonal differences in the location and number of pollution cluster, the

trajectory statistics is given for the total Pacific as well as northern and southern NH-Pacific separately.

**Fig. 4. Right panels. The very low value of the JJA 2001 data point coincides with MOPITT not acquiring data between May and August 2001. Similarly suspect points: DJF 2016 (no data acquired some days in the Feb-Mar period), DJF 2009 (no data acquired some days in the Jan-Feb period). Were periods with missing data accounted for? If not, then some results and conclusions may be invalid. Unclear what other results/conclusions could be affected by this issue.**

1) Please see our general comment at the beginning of the reply.

2) We decided to include the points in the figure, as the quantities are weighted, thus the values given in Fig. 4 are actually independent of the number of days/events considered. Though, apparently there are extraordinary few CO maxima events on the (rather few) days considered in JJA in 2001.

**Fig. 4 (left panels) What is the relationship between ENSO and the data points plotted? It is unclear from the text why ENSO is discussed.**

We included the ENSO index in Fig. 4 to get an idea if years with a strong deviation from the mean (or better from the trend shown by the regression line) are correlated with El Nino episodes as these are known to change transport patterns over the Pacific and therefore also pollution export from Asia.
→ We slightly rephrased the corresponding part of the text (p-10).

**Fig. 4: Could those trends be caused by CO emissions elsewhere? Do we see less clusters and/or less CO-maxima days through time in the region studied because other regions in the planet are "dominating" that (relative) top 2 %? Consider a hypothetical scenario with increasing summer CO fire emissions over N America during the 2000-2019 period. Under such scenario: 1) the number of daily observations at the top 2 % would, increasingly, be found over/near N America and 2) conversely, the number of daily observations at the top 2 % over the N Pacific region would decrease during the same period.**

As we discuss in section 3.1 (p.10, l.205-210) this could indeed be the case.

We composed the same statistics as shown if fig. 4 also for other regions where we detect the 2%-grid points most often (i.e. Central Africa, southern Africa, South America). We included these plots at the end of this reply.

**Fig. 8b seems to indicate that a very large proportion of the trajectories (most of the trajectories in some cases; e.g., Russia DJF) initiate over the ocean. Does that mean that the CO source is at the ocean? If so, please explain. What's the relevance/significance of trajectories initiating over land versus over ocean?**

Fig. 8b refers to the *uplift* region. Not the *source* region.

***Fig 1, top left panel. The sharp contrast in average CO between land and ocean is suspect.***

After rephrasing section 3 we removed Fig. 1 from the manuscript.

The color code of the figure might have been misleading. This figure and the shown quantity is not used for any analysis and we had only included it into the manuscript to give a reader unfamiliar with the global CO distribution a vague orientation.

***Figures need to be arranged in the order in which they are mentioned in the text. For example, Fig. 3 is mentioned in the text after Fig. 1 but before Fig. 2.***

We mention Fig.3 once before Fig.2 but not to discuss the science shown (which is done after Fig.2 is discussed) but to show generally the region which is included in our analysis – which can best be seen in Fig.3. Therefore we think it is justified to deviate from the standard to mention the figures in the order of their numbering.

***- The legends of panels 8.d, 8.e, and 8.f include a "NE-Asia" class in white. Please show its boundaries in panel 8.a.***

As we only consider continental source regions, the land/sea boundary marks the boundary of the NE-Asia source region (the same for all other source regions).

***- Revise standard deviation representation in Fig. 8, for consistency. For clarity, consider plotting +- 1 standard deviation lines (not just -1 st. dev. lines) in different colors and/or with an horizontal offset to avoid overlaps.***

We mention in the figure caption that the standard deviation is only plotted in the negative direction. This was done by purpose as the standard deviation is the same in positive and negative direction and the bar charts already contain a lot of information. We considered the plot as confusing if more lines were included.

**→ We revised the manuscript carefully following the suggestions below.**

- Fig. 1. Please label panels. Same comment applies to other figures.

- Maps lack latitude and longitude labels (Fig. 1, 2, 3, 5, 7, 8.a).

- Fig. 8 caption: please explain yellow area in panel 8.a, not represented by color or name in any other panel. The explanation is in the text (lines 275-276); please include in caption too, for clarity.

- Fig 8.c: white bars. According to the text (lines 133-134) those represent the "rest" class. Please add label to panel 8.c for clarity. Rename "rest" to "other" or similar.

- Fig. 8.a. caption: For clarity and to avoid language issues, please consider rewording to, for example: "Figure 8. Summary of trajectory analysis results. (a) Source regions: China (green), NE-Asia (white), India (red), SE-Asia (blue), Russia (gray), and ??? (yellow). (b) Source surface type (land, ocean) per region and season. (c) Uplift type (warm conveyor belt, frontal system, other) per

region and season. (d) Source region per season. (e) Same for the NE-Pacific region only. (f) Same for the Southern Pacific region only."

- Expressions such as "surprisingly high" (line 7), "extraordinary [sic] high" (165, 172), "extraordinary [sic] large" (212) should be avoided. Objective, quantitative statements should be used instead.

- line 58: "North America"

- line 61: Please consider rewording to "However, in that particular case study" or similar. There are several other cases where "Though" is used at the beginning of a sentence (line 22, 144, 151, 163, 179, 200, 204, 219, 254, 269, 272, 299, 304). Please reword.

- line 125: Please reword "tends to rather underestimate than overestimate" to " tends to underestimate rather than overestimate".

- line 154: "top and centre rows".
- line 234: "both clusters".
- line 277: "The two case studies indicate that pollution".

**Central Africa**

(A) Daily mean number of:
   COmax grid points

(B) Daily mean number of:
   COmax cluster

(C) Season mean number of:
   COmax days

[Figure]

**South Africa**

(A) Daily mean number of:
   COmax grid points

(B) Daily mean number of:
   COmax cluster

(C) Season mean number of:
   COmax days

[Figure]

**Central Africa**

(A) Daily mean number of:
   COmax grid points

(B) Daily mean number of:
   COmax cluster

(C) Season mean number of:
   COmax days

---

## Referee Report (RR1)

Review of "Contribution of Asian emissions to upper tropospheric CO over the remote Pacific" by Smoydzin and Hoor

Smoydzin and Hoor presented a well-written paper which I enjoyed reviewing. Overall, the quality of the content is high. The authors analyzed space-borne MOPITT CO data in an innovative way, which leads to multiple new and interesting results. The study period covers 20 years (2000-2019) (the backward trajectory simulation covers 2000-2018) so this is a large undertaking. With analysis based on daily CO values for this long period, the statistic results are robust. I recommend this paper to be accepted.

I have some minor comments for the authors to consider, mostly for clarification.

L56, "LRT events" is defined as "This AOD distribution is log-normal and Luan and Jaeglé (2013) choose the top 20% days in the frequency distribution as LRT events". It is unclear how the days are connected to the events.

L75, the authors selected 400 hPa, a level at the middle to upper troposphere where MOPITT CO data are less biased than at the other levels. Nevertheless, the signal at this level may still be contaminated by signals at the other levels. It is informative to show some information on the vertical sensitivity of MOPITT data over the NH-Pacific, in the main manuscript or Supplement, for example, in term of the vertical distribution of the average kernels over the NH-Pacific. Are both daytime data and nighttime data used in the analysis?

L80, "A global coverage" can be changed to "A near-complete global coverage".

L139, "Though, the linearity (and therefore the trend) of the data set is rather weak."? The trend appears statistically significant. Please check.

L155, which ENSO index is used? When correlate the ENSO index with CO mixing ratios, is the lag set as 0?

L171, "16 days" would be too long and the data from the last few days may not be reliable. Please provide an justification for this selection

L176 and 254, please change "850hpa" to "850 hPa". Please leave a space between a value and its unit throughout the text.

L302, "We identified"? To be consistent with the rest of the sentences, "We identify" is better.

Fig. 8a. There are two areas in white. Change one of them in a different color.

---

## Author Response (AR2)

Review of "Contribution of Asian emissions to upper tropospheric CO over the remote Pacific"
by Smoydzin and Hoor
Smoydzin and Hoor presented a well-written paper which I enjoyed reviewing. Overall, the
quality of the content is high. The authors analyzed space-borne MOPITT CO data in an
innovative way, which leads to multiple new and interesting results. The study period covers 20
years (2000-2019) (the backward trajectory simulation covers 2000-2018) so this is a large
undertaking. With analysis based on daily CO values for this long period, the statistic results are
robust. I recommend this paper to be accepted.

I have some minor comments for the authors to consider, mostly for clarification.

**L56, "LRT events" is defined as "This AOD distribution is log-normal and Luan and Jaeglé
(2013) choose the top 20% days in the frequency distribution as LRT events". It is unclear
how the days are connected to the events.**
This is explained in detail in Luan and Jaegle´ (2013). They choose those 20% days of their study
period  having the highest AOD values and select them as LRT events of pollution from Asia over
the Pacific.

**L75, the authors selected 400 hPa, a level at the middle to upper troposphere where MOPITT
COdata are less biased than at the other levels. Nevertheless, the signal at this level may still
be contaminated by signals at the other levels. It is informative to show some information on
the vertical sensitivity of MOPITT data over the NH-Pacific, in the main manuscript or
Supplement,for example, in term of the vertical distribution of the average kernels over the
NH-Pacific. Are both daytime data and nighttime data used in the analysis?**
We added a supplement to the manuscript showing the averaging kernels for our study region and
added a sentence in section 2.1 referring to the supplement

**L80, "A global coverage" can be changed to "A near-complete global coverage".**
We changed this line.

**L139, "Though, the linearity (and therefore the trend) of the data set is rather weak."? The
trend appears statistically significant. Please check.**
We rephrased this line:
In particular, during winter time the decrease of CO-maxima days is statistically significant.
We find CO-maxima on ~80% of all days in 2001 while less than 60% of all days in DJF are
selected in 2019 (Fig. 4(C) DJF). However, the correlation of the data points is
rather weak (r<0.5) showing a high interannual variability.

**L155, which ENSO index is used? When correlate the ENSO index with CO mixing ratios, is
the lag set as 0?**
(1) Yes!
(2) We use the Oceanic Niño Index.
We realised that the link to the web page was incomplete and we corrected it in the caption of
Fig. 4 (https://origin.cpc.ncep.noaa.gov/products/analysis_monitoring/ensostuff/ONI_v5.php)

**L171, "16 days" would be too long and the data from the last few days may not be reliable.
Please provide an justification for this selection**
This is indeed true. We performed the simulations for such a long time period simply to ensure that
we capture all source regions. Trajectories from India need longer to reach the eastern NH-Pacific
than those from China.

In the manuscript, we only mention the average residence time of the trajectories in the free troposphere (End of section 3) which is on average less than 7 days for all trajectories (apart from those from India). For all trajectories included in the statistics, the considered time period is on average less than 10 days (time between crossing a CO emission region and reaching the CO maxima) again with trajectories from India being an exception.

**L176 and 254, please change "850hpa" to "850 hPa". Please leave a space between a value and its unit throughout the text.**
We changed this.

**L302, "We identified"? To be consistent with the rest of the sentences, "We identify" is better.**
We changed this.

**Fig. 8a. There are two areas in white. Change one of them in a different color.**
We slightly changed this plot.